# Capital Intensity and Labour Productivity in Waste Companies

**Martina Novotná** , **Ivana Faltová Leitmanová** , **Jiří Alina** * and **Tomáš Volek**

Department of Applied Economy and Economics, Faculty of Economics,
University of South Bohemia in České Budějovice, Studentská 13, 370 05 České Budějovice, Czech Republic;
novotna@ef.jcu.cz (M.N.); leitman@ef.jcu.cz (I.F.L.); volek@ef.jcu.cz (T.V.)
* Correspondence: jalina@ef.jcu.cz; Tel.: +420-3877-72500

**Abstract:** At first glance, it might seem that the economic aspects of sustainability in terms of waste management have resolved themselves already in areas of activity. In reality, however, companies in this area also need to address how to ensure their future operations. The primary priority for companies in the area of waste disposal is to provide efficient collection, sorting, and recycling, effectively using company resources. The goal of this paper was to explore the relation between capital intensity and the productivity of labour in companies in the waste sector in the countries of the Visegrad Group (V4), and consequently, to define the bonds among economic indicators in the form of the economic normal. The study used data from 875 enterprises from the V4 countries, which were divided into categories according to the development of capital intensity and labour productivity. This study found that companies mainly implement modest investment development, which was characterised by the high effectiveness of capital usage, diminishing labour productivity, low labour endowment, but at the same time, increasing profitability. The reason for the labour productivity decrease was due to the growing proportional cost of labour. This trend was typical for most of the large-sized and middle-sized companies, whereas for most small companies, there was a dominant severe capital development with decreasing labour productivity and relatively high profitability of incomes. The smallest representation takes companies with capital-intensive development with the positive development of all monitored economic indicators.

**Keywords:** waste; capital–labour ratio; labour productivity; economic normal

## 1. Introduction

Each sector has certain specifics given to it by its activity, and this is also valid for the waste sector. On the one hand, it is possible to expect an expressive growth in investment in the context of increasing waste volumes and subsequently, tremendous pressure for waste collection and recycling. On the other hand, there are questions about whether companies should invest in new modern technologies and whether they use the investments sufficiently effectively; that is, the effectivity of production factors increases labour productivity, which contributes to the increasing efficiency of a company (increasing company profitability).

This effectiveness can be affected by the concrete economic activity of a company, company size, as well as by the various conditions in particular Visegrad Group (V4) countries. The companies were chosen according to their prevailing activities (381 waste collection companies, 382 waste disposal and treatment companies, 383 materials recovery companies) , but they may operate all mentioned activities from another branch. It is necessary to note that the paper is focused on the post-communist countries of Central and Eastern Europe, concretely, the Czech Republic (CZ), Poland (PL), Slovakia (SK), and Hungary (HU), that passed through significant social, economic, and especially structural

changes. This growth affected not only increasing waste production but also its collection, processing, removal, and reuse.

The purpose of the article is to explore the relation between capital–labour (c.l.) and company labour productivity and other connected indicators in the waste sector of the Visegrad Group countries. There was an intention to categorise companies based on an empirical study of accounting data and to ascertain the dynamics and bindings of economic characteristics in the form of the so-called "economic normal" typical for creating company groups.

Above all, the dynamics of the c.l. ratio indicator were analysed in relation to labour productivity, which differs in companies primarily due to the fact of whether companies engage in investing in intensive development (high capital intensity), or investing in modest development (low capital intensity). The experiential study included accounting data from 785 companies in the waste sector and materials recovery from the Visegrad Group countries. The direction of capital intensity has a consequent influence on company profitability. The main contribution from this study is a more in-depth analysis of capital intensity and related indicators in the area of waste activities, which has not been complexly analysed for all types of waste before. This analysis can be useful for policymakers during instrument creation and economic policy provisions for this sector.

## 2. Literature Review

Although companies are established mostly for the purpose of independent and repeatable activities that generate profit, they also have so-called "alternative objectives". Company activities have already been roofed over by so-called "responsible production and consumption" since the beginning of the millennium (and more recently, since 2015), at least in the context of the Sustainable Development Goals (SDGs), [1]. Responsible production seems to be incredibly intensive in the case of waste sector companies [2]. Despite the differentiated motivation of companies, it is impossible to resign ourselves to the necessity of tracking the effectiveness of these activities, mainly when they use rare, scarce, and often non-renewable resources [3–5]. The chosen financial indicators represent one of the tools that can be used for assessing effectiveness [6].

The effort to affect the sustainable perspective in the context of limited input was considered when authors analysed the sector, in which can be found a direct linkage on circular economy processes, which are focused on waste arrangements for other uses. This approach is gaining interest in the context of particular waste groups, and also for company objectives [7–10].

Although the comparison of countries, according to different criteria, is a common and usual method, there are only a few studies that use a country's business data for comparison [11–14]. Additionally, there are not commonly studies that carry out these comparisons with a focus on the waste sector as well. Despite this, it is possible to find research papers that focus on efficiency and productivity in relation to a particular type of waste [15–17].

The fundamental theoretical starting point for research is the neo-classic model production function, and consequently, its microeconomics application. The Cobb–Douglas production function has been in economic science since 1928. It started in 1928 when Charles Cobb and Paul Douglas [18] published their seminal paper in which they expressed a connection between the volume of inputs and the output of physical production in American manufacturing. The connection can be expressed as the following equation:

$$Y = f(L, K) = AL^kK^{1-k} \tag{1}$$

where

- Y is the output,
- A is the level of technology,
- K is the capital stock,
- L is the quantity of labour.

The coefficients k and (1 − k) are both between zero and unity. They represent the relative importance of the two factors in production [19].

Nowadays, a vast number of researchers, scientists, and academic workers are doing research activities that use the basic Cobb–Douglas function principles. For example, Ref [20] offers an alternative approach for solving and minimising the costs of production. The Cobb–Douglas production function was analysed for modelling production at the sector level for an extended period [21,22].

The effort to explain the relationship between input and output led to partial modifications and adjustments of the original form of the theoretic model of the Cobb–Douglas production function. Ref [23–25] calculated and used, in logarithmic terms, the index numbers of fixed capital (K), the total number of workers employed (L) in U.S. manufacturing, and physical production in manufacturing (Y). Here are some examples of state-of-the-art approaches and the use of the Cobb–Douglas production function: Authors [26] considered stochastic restrictions for the model, based on auxiliary information to improve its predictive ability of the Cobb–Douglas function. Other authors [27] estimated parameters for the function with the maximum likelihood method, which allows comparing the standard case with a flexible framework that provides a robust estimation of parameters based on the Student-t case. Authors [28] directly applied a Cobb–Douglas symmetric plane using the index of surface openness, which is used in geography, and successfully identified it. Research on the dynamic relationship between natural gas consumption and economic growth was done by [29], and they used the neoclassical economic growth theory, the expanded production function. Authors [30] used empirical looks into an analytical justification of using the Cobb–Douglas production model in order to estimate and test the coefficients of the production inputs for each of the chosen manufacturing industries using annual industrial statistical data.

Capital and sources of production factors are key variables for our paper. An example of this may be a paper written by [31], where the authors also analysed the supply and demand forces in capital markets and the cost and availability of capital funds as a critical phenomenon of GNP (gross national product) and income predictions.

With insufficient home country resources and a possibility for better resource appreciation leads to, among others, a volume and structure of the direct foreign investment. FDI (foreign direct investment) is considered to be an investment made by firms in one state into economic interests situated in another country. In general, FDI happens when the investor establishes foreign economic activity of foreign business assets in a foreign firm or company. Ref [32] examined research and development spillovers from foreign direct investment (FDI) in terms of both horizontal and vertical involvement (forward and backward).

In the monitored countries, foreign direct investment represented a significant factor for economic growth that culminated especially at the turn of the millennium. Their impact was influenced by institutional heterogeneity [33], the structure of the economy [34], and by social factors [35]. The effects of foreign direct investment on the development of companies and the economy examined [36]. Foreign direct investment is the primary source of capital for corporate restructuring.

We assume the investment is an asset bought by a company with the main target of generating income for the firm. The investment is the specially purchased product that is not consumed but is used in the future to create other products. The polish study considers that economic quality is being developed gradually and mainly concerns the most developed countries [37]. It is significant to recognise that the modern economy is different than the traditional (in which capital, labour, raw materials, adequate infrastructure, efficient transport, and proper organisation were the decisive factors) and the current information and knowledge-based ones. Modern technologies, highly qualified workers, universities, research and development, information and communication technology (ICT), as well as state pro-innovation policies, gradually become the basis for economic development. It is a complicated, long-term and regionally varied process. There is no doubt that we are dealing with fundamental changes, and human capital plays a key role in their intellectual and social aspects [38].

At the same time, it is necessary to take into account the considerable indeterminateness of a possible development scenario that is reflected in company decision-making. It is also essential to search for ways about how to, at least partly, eliminate the indeterminateness [39–41].

Models of firm behaviour in conditions of uncertainty show that the choice of firm inputs and cost functions can be affected by uncertainty. Research in US manufacturingexamined the firm's input choices in a model where capital is selected in advance (initially), but both labour and output are determined by demand [42]. The authors show that when there is a possibility of substitution between capital and labour, an increase in demand in a situation of uncertainty can lead to lower optimal share capital for companies and greater labour use. Firms can use less capital and operate with a lower (less efficient) capital–labour ratio.

Capital intensity is defined by [43] as a ratio of capital to labour input used in an economy, sector, or company. Quite often, the term *capital intensity* is replaced by the capital–labour ratio (c.l. ratio). On the company level, not only is the general capital intensity observed but also the capital intensity of company investments [44]. The dynamics and amount of capital intensity depend on the kind of sector [45]. The paper by [46], on New Zealand and the UK, shows an expressive difference in capital intensity in the areas of manufacturing and services, to which the analysed waste sector belongs. Unique studies that deal with capital intensity in the waste sector do not exist. Labour productivity has a close binding on capital intensity. The labour productivity in companies measures the effectiveness of labour use in the company given to technology and capital endowment. The paper by [47] on a sample of Swedish companies confirmed a link between labour productivity change and a change of capital intensity.

Variously defined ratio indicators can measure the level and dynamics of capital intensity. For an examination and assessment of the dynamics of the fixed assets turnover (FAT), it is useful to spread out this detector on labour productivity (LP) and the capital–labour ratio (c.l. ratio), which can be written as the following equation:

$$FAT = \frac{R}{(TFA + IFA)} = \frac{R}{CE} : \frac{(TFA + IFA)}{CE} = \text{LP} : \text{c.l.ratio} \qquad (2)$$

where

- (*TFA* + *IFA*) are tangible + intangible fixed assets,
- *R* is operating revenues,
- *CE* is Cost of employees,
- LP is labour productivity,
- c.l. ratio is the capital–labour ratio,
- *FAT* is the Fixed Assets Turnover.

The same relations are also valid for indexes of these indicators:

$$i_{FAT} = \frac{i_{LP}}{i_{c.l.ratio}} \qquad (3)$$

If the index of fixed assets turnover is equal to one, it is evident that the index labour productivity is equal to the index c.l. ratio and labour productivity growth proportionately to capital intensity. This relation is characterised by the neutral capital development of the company and the company is achieving an effect only from the extensive expansion of manufacturing capacity.

If $i_{FAT} > 1$ then $i_{LP} > i_{c.l.ratio}$. It is a *modest* company *investment development*.

This case leads to increased productive utilisation of fixed assets.

If $i_{FAT} < 1$ then $i_{LP} < i_{c.l.ratio}$. It is an *intensive* company *investment development*.

In this case the productivity of labour grows slower than the c.l. ratio.

Slower growth in labour productivity than c.l.ratio was related to growth of unit cost for production in agriculture [48]. For the analysis of the relationship between indices for the chosen indicators,

it is useful to express indices through inequation. These inequations can be used mainly in business economics, especially for assessing the qualities of the company economy. From inequations on a business level, Hoffman created [49] the so-called "economic normal". The economic normal was defined by [49] and consequently by [50] as a system of inequalities that enables them to indicate a positive development of economic indicators (in the context of the indicators' relationships).

For another way to measure the efficiency of the waste sector companies it is possible to use a modified DEA (Data Envelopment Analysis) data panel method [51] or an order-M data panel frontier analysis, which provides more robust results [52,53].

The structure of this paper is as follows: In Section 2, the authors look at capital intensity in the context of macroeconomics, and the microeconomics approach is explained. In Section 3, the methodological procedure and used data are presented. In Section 4, the main empirical results and observations are presented and discussed. Section 5 summarises the results and points out an area for further research.

## 3. Materials and Methods

This paper focuses on the analysis of heterogeneous companies from the viewpoint of capital intensity in the year 2018, in comparison with the year 2013. 'The research question the authors solved was whether it is possible to generalise partial findings and create an "economic normal" at the level of companies with specifics for the waste sector. There was an empirical study on about 875 companies in countries of the Visegrad Group (V4) (Czech Republic (CZ), Hungary (HU), Poland (PL), and Slovakia (SK)), ranked into sectors according to the NACE (Nomenclature statistique des activités économiques dans la Communauté européenne) Section E: water supply; sewerage, waste management and remediation activities, waste collection, treatment and disposal activities, and materials recovery. Financial data about the monitored companies were used from the database AMADEUS.

The analysis was at first focused on the examination of the difference of levels of capital intensity, namely among particular countries, particular subsections of waste activities, and the size of companies' groups. The hypothesis of the level of the capital–labour ratio in companies was verified by using the ANOVA test [54]. This test allowed us to test the effect of multiple factors on a variable. The variable explained was the c.l. ratio, the explanatory variables, referred to as factors, were the size of groups (small, medium, and large, by the classification of [55], based on the number of employees, turnover, and balance sheet total) and Visegrad countries (Czech Republic, Slovakia, Hungary, and Poland) and NACE code (381 waste collection, 382 waste treatment and disposal, 383 materials recovery). We tested the hypothesis (see below) about the so-called main effects of factors, that is., hypotheses that the effects of all levels of a given factor (regardless of the level of the second factor) are zero.

H: $X_1 = X_2 = \ldots = X_k = 0$ X (groups of companies according to size) $= 1, \ldots ,3$;
respectively,
H: $Y_1 = Y_2 = \ldots = Y_k = 0$, Y (groups of companies by V4 countries) $= 1, \ldots ,4$;
respectively,
H: $Z_1 = Z_2 = \ldots = Z_k = 0$, Z (groups of companies by economic activities) $= 1, \ldots , 3$.

This means the hypothesis that the magnitude of the effect of a change in the level of one factor does not depend on the specific level of the other factor.

Dynamics of the capital intensity indicator were under consideration in relation to the dynamics of the labour productivity indicator, and consequently, to other indicators (Table 1). Table 1 provides a definition of the indicators and descriptive statistics found on the sites of 785 enterprises and Table 2 represents frequencies of analysed companies by size and country to clearly show the companies in countris and their sizes.

**Table 1.** Used indicators and description statistics.

| Short Form | Indicator Name | Definition | 2013 | | 2018 | |
|---|---|---|---|---|---|---|
| | | | Average | Median | Average | Median |
| c.l. ratio | Capital–Labour Ratio in Euro | Tangible and intangible fixed assets/Costs of employees | 3.832 | 1.770 | 3.410 | 1.645 |
| LP | Labour Productivity in Euro | Operating revenues/Costs of employees | 6.049 | 4.683 | 5.582 | 4.293 |
| FAT | Fixed Assets Turnover in Euro | Operating revenues/Tangible + intangible fixed assets/ | 1.579 | 2.927 | 1.637 | 2.808 |
| ROS | Return on Sales in Euro | Profit/loss before Taxes/Operating revenues | 0.046 | 0.033 | 0.050 | 0.035 |
| LCR | Labour Cost Ratio in Euro | Costs of employees/Operating revenues | 0.165 | 0.214 | 0.179 | 0.233 |

Note: Number of observations: 785, Data from BALANCE SHEET and PROFIT/LOSS ACCOUNT; Source: Authors' calculations.

**Table 2.** Frequencies of analysed companies by size and country.

| Country | Size | | | Total |
|---|---|---|---|---|
| | Large | Medium | Small | |
| Czech Republic (CZ) | 10 | 74 | 97 | 181 |
| Hungary (HU) | 9 | 73 | 138 | 220 |
| Poland (PL) | 21 | 140 | 111 | 272 |
| Slovakia (SK) | 4 | 34 | 74 | 112 |
| Total | 44 | 321 | 420 | 785 |

Source: Authors' calculations.

The analysis of dynamics of the labour productivity and capital intensity indicators started with observing connections among economic indicators, which can be calculated using the following equation:

$$i_{FAT} = \frac{i_{LP}}{i_{c.l.ratio}} \ i.e., \ \frac{FAT_{2018}}{FAT_{2013}} = \frac{\frac{LP_{2018}}{LP_{2013}}}{\frac{c.l.ratio_{2018}}{c.l.ratio_{2013}}} = \frac{\frac{LP_{2018}}{c.l.ratio_{2018}}}{\frac{LP_{2013}}{c.l.ratio_{2013}}} \tag{4}$$

The dynamics of the indicators of labour productivity and the c.l. ratio were the foundation for the construction of four quadrants (four groups of companies); see Figure 1:

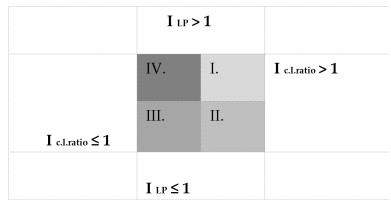

**Figure 1.** Labour productivity and capital intensity relationship.

Quadrant I: $I_{c.l.\ ratio} > 1 \wedge I_{LP} > 1$;

Quadrant II: $I_{c.l.\ ratio} > 1 \wedge I_{LP} \leq 1$;

Quadrant III: $I_{c.l.\ ratio} \leq 1 \wedge I_{LP} \leq 1$;

Quadrant IV: $I_{c.l.\ ratio} \leq 1 \wedge I_{LP} > 1$.

The levels and dynamics of all indicators in individual quadrants were examined.

## 4. Results and Discussion

The framework for single partial findings is given by the different structures of waste activities (Section E.38, according to the NACE classification), in particular V4 countries in the monitored period. The share of companies was predominately from the waste collection section (E.38.1) in the Czech Republic. On the other side, In the Czech Republic the share of companies from the waste treatment and disposal section (E.38.2) was the lowest among monitored countries. Conversely, the common trend of all monitored countries was the increasing share of companies from the materials recovery section (E.38.3). There was a growth of the share of companies in this area, and this occurred even when there was a drop in their absolute number. The creation of value-added in Section E.38 was not achieved by any of the monitored countries, not even at half the level of EU countries in total. The size of resource productivity (the GDP generated per unit of resources—labour or capital—used by the economy) was at half the level in these countries as compared with the average of the EU countries. Even Poland reached only 30% of level resource productivity in comparison with the EU average [56]. The growth of this indicator was positive, which was also affected by EU policy [57]. From the viewpoint structure of NACE Section 38, the waste activities situation is not incidental in the monitored countries. Waste collection predominates, except in Slovakia (381), where section waste treatment and disposal (Section E.38.2) predominates [56]. The share of companies that are concerned with materials recovery (Section E.38.3) is lower in comparison with the EU. Analysis by [58] focused on recycling in the EU and points out the great impact by low taxation from waste disposal at the level of recycling at Slovakia, Poland, and the Czech Republic.

Companies in Poland had the highest level of c.l. ratio values. Conversely, the lowest level was found at Czech companies. We found the biggest capital–labour ratio, according to expectations, at large companies, where, at the same time, the largest drop of this indicator happened in the year 2018, namely at 83% of the level from the year 2013, during which time high variability was recorded. The reason for the drop that occurred in all size categories was the same, namely the overgrowth of personnel expenses compared with capital growth. From a departmental viewpoint, the largest capital endowment is owned by companies focused on waste removal (382). The companies focused on recycling (383) have a high variability of values.

The c.l. ratio level and differentness of its value in the last monitored year (2018) were analysed by ANOVA (An F statistic is a value which is calculated by ANOVA test or a regression analysis to find out whether the means between two statistical samples are significantly different, the p value is determined by the F statistic and can be used for the probability of results ). ANOVA tests (Figures 2–4) show the results of the three-factor analysis of variance. The analysis showed that companies divided by size do not affect the level of capital intensity (the effect of the factor is insignificant, $p > 0.05$). The influence of the second factor, that is, the group of enterprises according to countries, is statistically significant ($p < 0.05$) (i.e., the influence of the country within the V4 on the level of capital intensity was proved). The influence of the third factor, that is, the group of enterprises according to the NACE code, is statistically significant ($p < 0.05$) too.

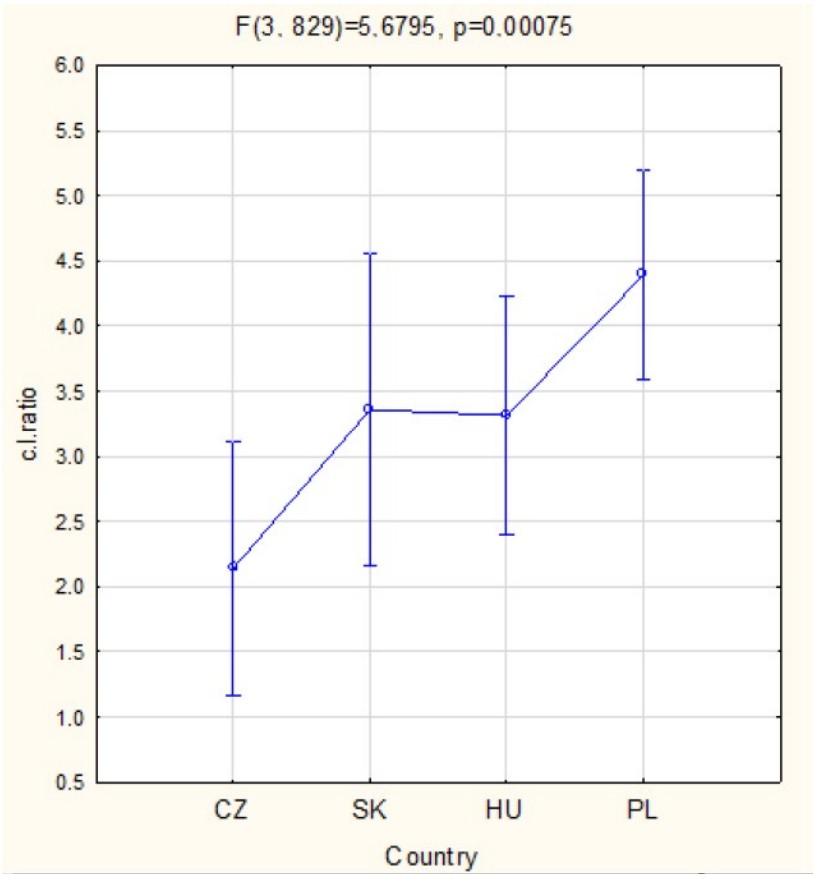

**Figure 2.** ANOVA tests of the three-factor analysis of variance (Country). Source: Authors' calculations.

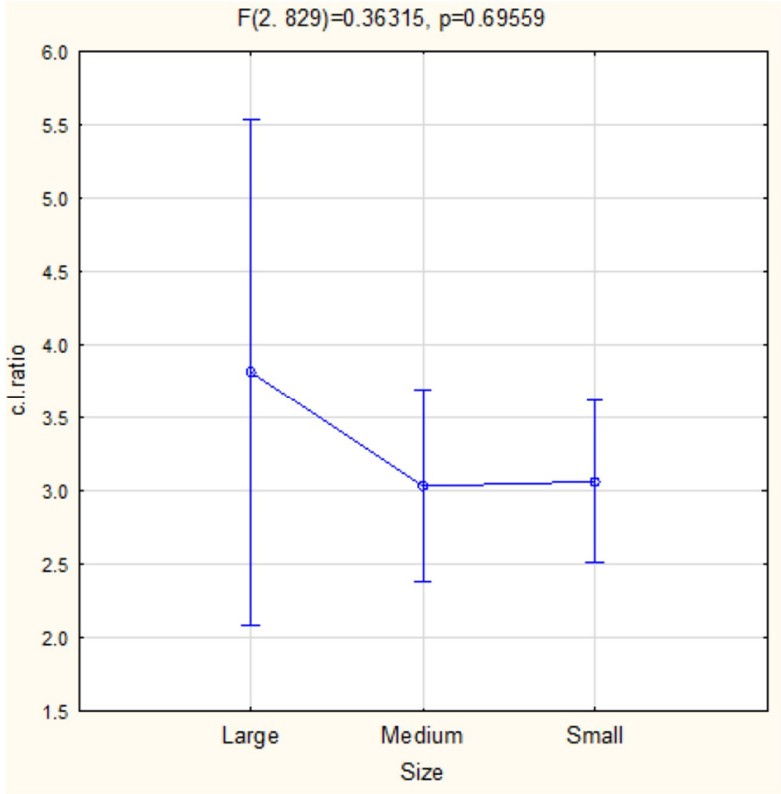

**Figure 3.** ANOVA tests of the three-factor analysis of variance (Size). Source: Authors' calculations.

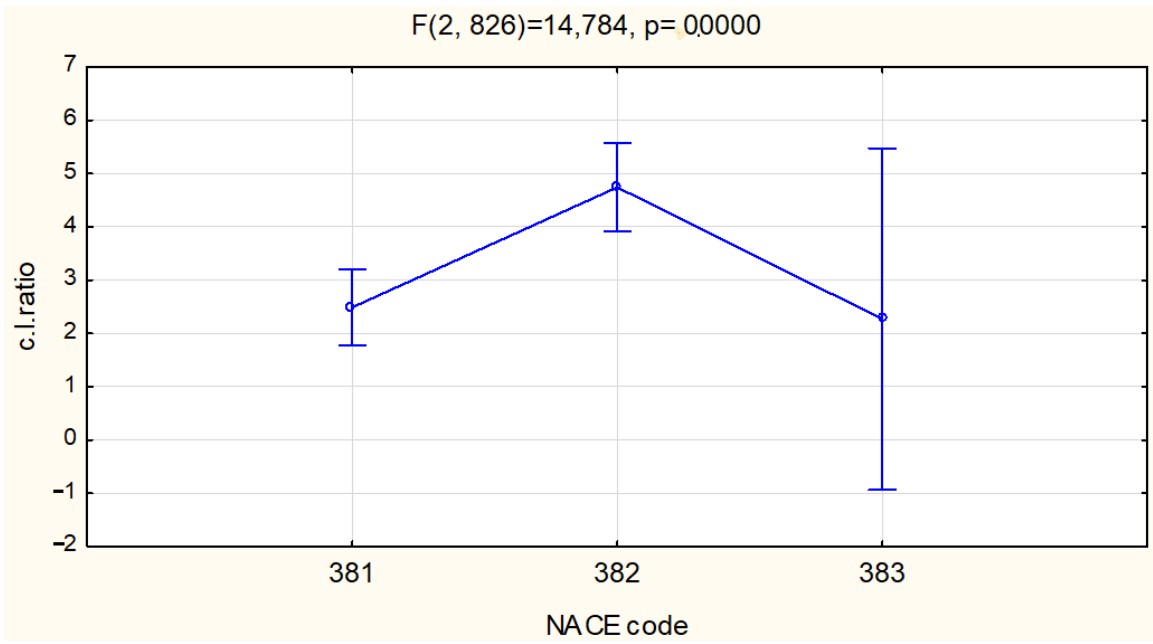

**Figure 4.** ANOVA tests of the three-factor analysis of variance (NACE code). Source: Authors' calculations.

The interval bars indicate 0.95 confidence intervals for Figures 2–4. Figure 5 represents the companies' layouts divided into four quadrants, based on the relationship between the c.l. ratio and labour productivity indicators (see methodology).

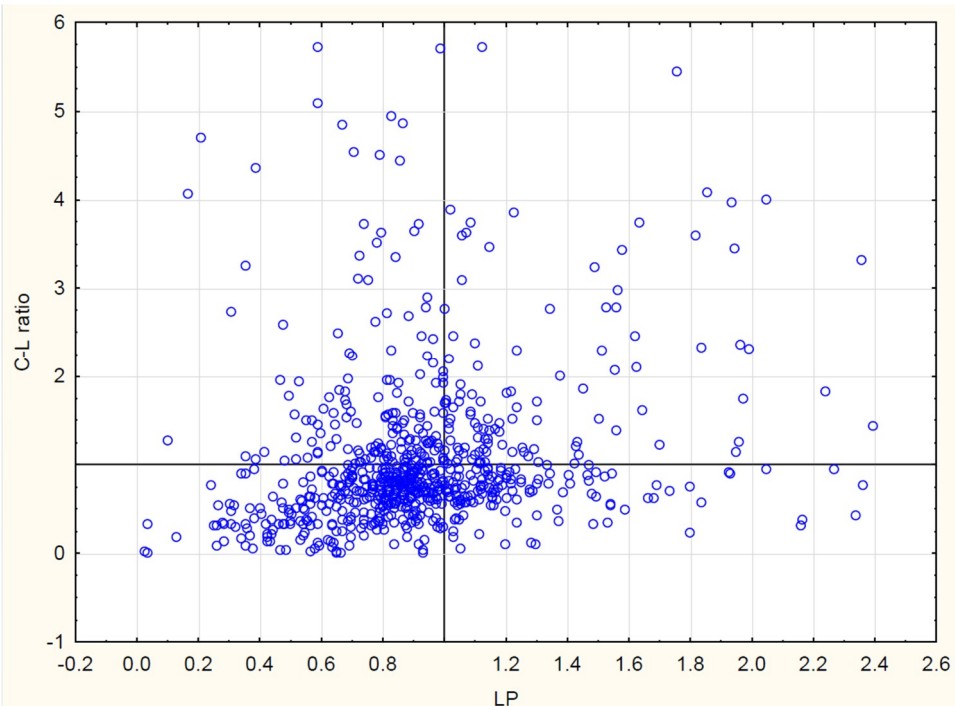

**Figure 5.** Division of enterprises. Source: Authors' calculations.

For companies classified by Figure 5, an average level of monitored indicators and their development in the year 2018 were consequently found in comparison with the year 2013 (Table 3). Numerically (how many times), Quadrant II is most represented; about 45% of all tracked companies were ranked in Quadrant II. On the contrary, the lowest representation is in Quadrant I, where we can find about 16% of the companies.

**Table 3.** Indicators by individual quadrants.

| Quadrant | Indicator | Average Value in | | Index |
|---|---|---|---|---|
| | | 2013 | 2018 | |
| Quadrant I (125 companies) | Return on Sales—ROS | 0.062 | 0.065 | 1.055 |
| | Labour Cost Ratio—LCR | 0.182 | 0.145 | 0.800 |
| | Labour Productivity—LP | 5.506 | 6.880 | 1.250 |
| | Capital–Labour ratio—c.l. ratio | 2.213 | 3.970 | 1.794 |
| | Fixed Assets Turnover—FAT | 2.488 | 1.733 | 0.696 |
| Quadrant II (170 companies) | Return on Sales—ROS | 0.066 | 0.050 | 0.759 |
| | Labour Cost Ratio—LCR | 0.171 | 0.213 | 1.244 |
| | Labour Productivity—LP | 5.840 | 4.696 | 0.804 |
| | Capital–Labour Ratio—c.l. ratio | 2.249 | 3.148 | 1.400 |
| | Fixed Assets Turnover—FAT | 2.597 | 1.492 | 0.574 |
| Quadrant III (350 companies) | Return on Sales—ROS | 0.041 | 0.051 | 1.225 |
| | Labour Cost Ratio—LCR | 0.152 | 0.190 | 1.256 |
| | Labour Productivity—LP | 6.594 | 5.250 | 0.796 |
| | Capital–Labour Ratio—c.l. ratio | 3.735 | 2.401 | 0.643 |
| | Fixed Assets Turnover—FAT | 1.766 | 2.187 | 1.238 |
| Quadrant IV (140 companies) | Return on Sales—ROS | 0.022 | 0.034 | 1.571 |
| | Labour Cost Ratio—LCR | 0.189 | 0.155 | 0.823 |
| | Labour Productivity—LP | 5.296 | 6.432 | 1.214 |
| | Capital–Labour Ratio—c.l. ratio | 7.596 | 6.212 | 0.818 |
| | Fixed Assets Turnover—FAT | 0.697 | 1.035 | 1.485 |

Source: Authors' calculations.

In Quadrant I, there were companies with capital-intensive development. Their fixed assets turnover in the monitored years slowed down. The reason could be the aggressive growth of fixed assets on the Euro cost of employees, that is, the c.l. ratio (the average growth was about 79.4%—the highest of all monitored quadrants). Labour productivity reached a high level, and this level even increased in the monitored period (a growth of 25%). At the same time, these companies reached the maximum level of profitability on average, which increased gently.

The companies in Quadrant II reached the level of profitability on average in the year 2013, comparable with companies in Quadrant I. Still, in the year 2018, their Return on Sales (ROS) decreased by almost 24%, and at the same time, a decrease in labour productivity happened. The reasons could be an investment into fixed assets, as the c.l. ratio indicator increased by 40%, but these fixed assets were either less productive or their usage was inadequate; in any event, it was with a time delay, which led to a lowering of labour productivity. The largest share of small companies can be found in this Quadrant (68.6% of all monitored small companies) and at the same time, the smallest share of large companies. We may then claim that during the monitored period, this development was typical for small companies in Section 38.

In Quadrant III there were companies that showed investing in intensive development, that is, their fixed assets turnover increased, but labour productivity declined. At the same time, the c.l. ratio declined, and there was a relative reduction of fixed assets conversely relative, exceeding that of personal expenses. Economic success, such as a company's profitability, depends on the influence of both production factors. The average value of ROS indicators in both years implies a lower level in comparison with Quadrant I and II, but this indicator grew (a growth of 22.5%, on average). The revenue capacity grew faster than the fixed assets capacity but slower than the cost of an employee. This development was typical for many companies in recent years when, across sectors, wages increased and often grew faster than labour productivity [59]. The largest share of large and middle-sized companies can be found in this Quadrant (54.5% of all large companies) and middle-sized companies (43% of all middle-sized companies). Here (in Quadrant III), there are no dominant companies from the perspective of country or activity (381, 382, and 383).

The companies in Quadrant IV can be characterised as modest, in terms of investment development, because the c.l. ratio declined. However, the usage of production factor labour increased because labour productivity increased. There was a relative saving of fixed assets and, at the same time, a relative saving on expenditure on employees. This variant was economically prosperous, but it is evident from the average values of monitored indicators that companies reached the lowest levels of profitability of revenues (ROS). Indeed, the low value of ROS in the year 2013 enabled a high percentage of growth in the year 2018 (a growth of 57%). Companies from Poland had the second-largest representation in this Quadrant, and this Quadrant was also predominately middle and large-sized companies.

From the analyses of the relationship between indicators, particular quadrants were created by an inequation, the so-called economic normal, which generalise these outcomes (Table 4).

**Table 4.** Economics normal for Quadrants.

| Quadrant | Inequalities | Investment Development |
|:---:|:---:|:---:|
| Quadrant I | $I_{c.l.ratio} > I_{LP} > I_{ROS} > 1 > I_{LCR} > I_{FAT}$ | Intensive |
| Quadrant II | $I_{c.l.ratio} > I_{LCR} > 1 > I_{LP} > I_{ROS} > I_{FAT}$ | Intensive |
| Quadrant III | $I_{LCR} > I_{FAT} > I_{ROS} > 1 > I_{LP} > I_{c.l.ratio}$ | Modest |
| Quadrant IV | $I_{ROS} > I_{FAT} > I_{LP} > 1 > I_{LCR} > I_{c.l.ratio}$ | Modest |

Source: Authors' calculation.

From the view of an attempt to increase investment intensity, which at the same time presents the growth of labour productivity and profitability, it is possible to recommend for companies (in the waste area) to apply the economic normal, which can be found in Quadrant I. This development expects a diminishing level of the labour cost ratio to be better achieved, especially at middle and large-sized companies, which can realise economies of scale. For small companies, development is typically characterised by the economic normal of Quadrant II, where a diminishing level of labour cost is difficult to achieve.

When companies choose the more likely modest investing development, which was the most frequent variant in the monitored period because there was growth in the cost of employees and therefore growth in the labour cost ratio, then investment intensity also decreases labour productivity, but may still grow profitability. Increase in the turnover rate highly affects the economisation (profitability incomes). Of course, when companies have rapid growth of labour productivity higher than the labour cost ratio (Quadrant IV), it can lead to a faster increase in profitability.

The empirical analysis shows that almost half of the companies in the period of economic growth (2013–2018) implemented investing in modest development (companies in Quadrant III). Quadrant III is characteristic for its high effectiveness of capital usage but low labour productivity, low labour endowment, and at the same time, lower profitability. The reason is the over-proportional growth of labour costs. Ref [50,60] claim that the excessive growth of personnel expenses in these countries is often due to low unemployment. For companies that choose modest investing development, it is possible to recommend an increase in the turnover rate of assets that positively influences the profitability of earnings; that is, in the case of a company where labour productivity grows faster than the labour cost ratio, there is a faster increase in earnings profitability.

On the other hand, companies in Quadrant I investing in intensive development have the smallest representation. The labour productivity and c.l. ratio growth in companies reached high profitability. In Hungarian SMEs was found that the growth of labour productivity leads to higher profitability of companies [61].

## 5. Conclusions

The waste sector is a part of every advanced economic country. The state is obliged, respecting social responsibility, to ensure not only the collection but also the removal of waste, as well as the reuse of waste in production (as a production factor).

The capital–labour ratio is one of the indicators influenced not only by the size of production in waste activities but also its dynamics. A study by [62] points out the influence of companies' ownership on their capital endowment. Other factors that influence the amount of capital may be EU funding policies. The study of Hungarian companies found that the productivity of labour in companies was not significantly affected by subsidies from the EU [63].

In this paper, we showed that the level of capital endowment in companies in the year 2018 is different in the V4 countries and is also different, according to their detailed activities (NACE). For the period under consideration, it was typical that the size of capital in this sector grew, but wage growth was markedly higher, which led to the fall of the c.l. ratio in the year 2018 for all countries.

The variants of development of monitored indicators (economic normal) seem to be optimal, from the viewpoint of the effort for implementing innovation and new technology in companies. The sense is that the variant has economic and environmental aspects as well. Innovated, thoughtful technology and investments connected to them represent an appropriate way to achieve sustainable development goals (SDGs) [64,65].

Regarding the size category of companies, modest investment development predominated, especially for large companies. For small-scale companies, intensive investment development predominated, which was characterised by higher capital investments on 1 EUR of personnel expenses and high labour productivity, but very low profitability.

Research limitations could be the fact that the authors analysed companies only during a period of economic growth in the monitored countries. Future research, therefore, should be focused on the monitored indicators in different phases of the business cycle. Furthermore, the impact of the applied instruments and measures of economic policies (pricing policy, subsidy policy) can be examined. Other limitations may be found in the sector inclusion of company activities. A limitation of the research may also be the fact that the authors did not consider the type of company ownership. The authors of the paper considered the basic concept that the aim of each enterprise is to create profit, regardless of the type of business ownership. Future research should include the types of business, and analysis should be done with other European Union countries; a comparative study should be performed.

**Author Contributions:** Conceptualization, I.F.L. and M.N.; methodology, M.N. and T.V.; writing – review and editing I.F.L., J.A. and M.N.; investigation and analysis, M.N., I.F.L. and T.V.; resources, I.F.L and J.A.; data curation, M.N. and T.V.; writing—original draft preparation, M.N., I.F.L. and J.A.; project administration, J.A.; funding acquisition, M.N. and T.V. All authors have read and agreed to the published version of the manuscript.

**Funding:** This research was funded by FACULTY OF ECONOMICS, UNIVERSITY OF SOUTH BOHEMIA, grant number IGS202005 and grant number IGS202008 and the APC was funded by grant number IGS202005 and grant number IGS202008.

**Conflicts of Interest:** The authors declare no conflict of interest.

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
