# Peer review of "Capital Intensity and Labour Productivity in Waste Companies"

_sustainability, doi:10.3390/su122410300_

Round 1
Reviewer 1 Report
- I didn’t find the answer in the paper about the statement which is written in the tile, “Waste Management ….. in firms?
- What is kind of waste management have been provided in compered firms in each country?
- How is relation in the size of companies, country in the field of method of waste management which is used.
- Answer for this question is expected according to the title of paper.
Author Response
- I didn’t find the answer in the paper about the statement which is written in the tile, “Waste Management ….. in firms?
The authors have changed the paper title to be better understandable. The new title is: “Capital intensity and labour productivity in waste companies” which fits more with the paper content and at the same time a reader can find answers for the given topic. The paper is not focused on the management of waste, but there is a focus on chosen economic indicators in companies in sector waste
- What is kind of waste management have been provided in compered firms in each country?
Authors didn't analyze firms according to structure of waste in particular country, but these analyzed firms in the sector waste in its entirety (according to classification NACE, code 38). The firm´s structure according to kind of waste management was used only for characterized selective file firm (Figure 4)
- How is relation in the size of companies, country in the field of method of waste management which is used.
Definite (clear) relationship between enterprise size and surveyed indicators wasn't proved (test ANOVA, Figure 1]. However the factor of enterprises sizes was taken into account in presented result (line 75 - 109).
- Answer for this question is expected according to the title of paper.
Title of paper was modified (changed). Authors dare assume, in regard to changed title, that the expected answers are in harmony with changed title.

Reviewer 2 Report
I think this manuscript is interesting but need important shortcomings that is necesary improve to publish it.
1. Literature review. This literature should be incluided: Cost efficiency in municipal solid waste service delivery. Alternative management forms in relation to local population size
G Pérez-López, D Prior, JL Zafra-Gómez… - European Journal of Operational Research, 2016; Temporal scale efficiency in DEA panel data estimations. An application to the solid waste disposal service in Spain G Perez-Lopez, JL Diego Prior, Zafra-Gomez Omega, 10.1016/j.omega.2017.03.005 Measuring the efficiency of public and private delivery forms: an application to the waste collection service using order-m data panel frontier analysis CM Campos-Alba, D la Higuera-Molina, J Emilio, G Pérez-López, ... Sustainability 11 (7), 2056. 2.- DEA model. It would be necesarry introduce Data Envelopment Analysis in the final model, in this way the analyss would be more robust and author/s could have one efficiency indicator.
Author Response
Reviewer 2
- Literature review. This literature should be included: Cost efficiency in municipal solid waste service delivery. Alternative management forms in relation to local population size
G Pérez-López, D Prior, JL Zafra-Gómez… - European Journal of Operational Research, 2016; Temporal scale efficiency in DEA panel data estimations. An application to the solid waste disposal service in Spain G Perez-Lopez, JL Diego Prior, Zafra-Gomez Omega, 10.1016/j.omega.2017.03.005 Measuring the efficiency of public and private delivery forms: an application to the waste collection service using order-m data panel frontier analysis CM Campos-Alba, D la Higuera-Molina, J Emilio, G Pérez-López, ... Sustainability 11 (7), 2056. 2.- DEA model. It would be necessary introduce Data Envelopment Analysis in the final model, in this way the analyses would be more robust and author/s could have one efficiency indicator.
The citations of publications, in accordance with recommendation, were added to Literature review. They present other possible approach to economic indicator analysis. The authors consider to use recommend methods in further and future research (line 192-194).
G Pérez-López, D Prior, JL Zafra-Gómez… - European Journal of Operational Research, 2016; Temporal scale efficiency in DEA panel data estimations. An application to the solid waste disposal service in Spain (line 193, [51],)
G Perez-Lopez, JL Diego Prior, Zafra-Gomez Omega, 10.1016/j.omega.2017.03.005 (line 194, [53])
Campos-Alba, D la Higuera-Molina, J Emilio, G Pérez-López, ... Sustainability 11 (7), 2056. 2.- DEA model. (line 194, [52])

Reviewer 3 Report
The paper studies how firms in the waste management sector manage their activities, namely concerning the balance between capital intensity and labor productivity (the study is empirical and focused on four Eastern Europe countries). The paper is written in a clear language and offers some results that might be considered of scientific value.
In my view, the paper would benefit from a revision in the following points:
1) There are some typos and grammatical errors that need correction. A careful last reading of the paper is required (just to give two examples (among many): there is an nonconformity between singular and plural in the first sentence of the abstract; there is an unnecessary word in the beginning of the conclusions section);
2) The introduction is too short and does not motivate well the analysis;
3) The attempt to explain the baseline theory through the presentation and explanation of the Cobb-Douglas production function and some ratios is weak. There is state-of-the-art literature at this level that could be surveyed;
4) The conclusions section is too long and disperse.
Author Response
The paper studies how firms in the waste management sector manage their activities, namely concerning the balance between capital intensity and labor productivity (the study is empirical and focused on four Eastern Europe countries). The paper is written in a clear language and offers some results that might be considered of scientific value.
In my view, the paper would benefit from a revision in the following points:
- There are some typos and grammatical errors that need correction. A careful last reading of the paper is required (just to give two examples (among many): there is an nonconformity between singular and plural in the first sentence of the abstract; there is an unnecessary word in the beginning of the conclusions section);
Some typos and grammatical errors were corrected (for example: abstract line 16, conclusions line 141 etc.)
- The introduction is too short and does not motivate well the analysis;
Introduction was improved. Added parts are (line 44-49, line 55-63):
- The attempt to explain the baseline theory through the presentation and explanation of the Cobb-Douglas production function and some ratios is weak. There is state-of-the-art literature at this level that could be surveyed;
The latest (state-of-the-art) knowledge about Cobb - Douglas production function theory was added into literature background research (review) – line 104-116.
- The conclusions section is too long and disperse.
The part „Conclusion “was revised and abridged. The results of empirical study are now presented more clearly and better. The paper contribution and at the same time its limits are evident. There is also outline for another research in this area.

Round 2
Reviewer 2 Report
I think this manuscript is ready to publish, I hope that future research the authors can develop new methodologies that add more robustness to their research
Reviewer 3 Report
The authors have made a relevant effort in improving the paper's contents, given the comments of the reviewers.